# Nutritional Status and Selected Adipokines in Children with Irritable Bowel Syndrome

**DOI:** 10.3390/nu14245282

**Published:** 2022-12-11

**Authors:** Wojciech Roczniak, Agnieszka Szymlak, Bogdan Mazur, Agata Chobot, Małgorzata Stojewska, Joanna Oświęcimska

**Affiliations:** 1Institute of Medicine, Jan Grodek State University in Sanok, ul. Mickiewicza 21, 38-500 Sanok, Poland; 2Department of General Paediatrics, University Hospital No 1 in Zabrze, Medical University of Silesia in Katowice, ul. 3 Maja 13-15, 41-800 Zabrze, Poland; 3Department of Microbiology and Immunology, Faculty of Medical Sciences in Zabrze, Medical University of Silesia in Katowice, ul. Jordana 19, 41-808 Zabrze, Poland; 4Department of Pediatrics, Institute of Medical Sciences, University of Opole, al. W.Witosa 26, 45-401 Opole, Poland; 5Department of Pediatrics, Faculty of Medical Sciences in Zabrze, Medical University of Silesia in Katowice, ul. 3-Maja 13-15, 41-800 Zabrze, Poland

**Keywords:** irritable bowel syndrome, leptin, adiponectin, chemerin, omentin-1, children

## Abstract

Background: The aim of this study was to assess the nutritional status and serum concentrations of adipokines in children with irritable bowel syndrome (IBS) and healthy controls. We also sought to evaluate their relation to metabolic parameters. Methods: We studied 33 IBS patients (11 girls, 22 boys) aged 5–17 years and 30 healthy age-matched controls (11 girls, 19 boys). The analysis included anthropometric measurements, body composition parameter measurements using bioimpedance, and biochemical tests and measurements of serum concentrations of leptin, adiponectin, chemerin, and omentin-1. Results: The results of the anthropometric measurements were comparable between the patients and the controls. The patients had higher triglycerides, HOMA-IRs, and chemerin concentrations than the healthy subjects. The HDL cholesterol and omentin-1 levels were lower than in the controls. Leptin and adiponectin did not differ significantly between the groups. An analysis of the receiver operator curves (ROCs) showed that serum concentrations of chemerin ≥ 232.8 ng/mL had 30% sensitivity and 87% specificity when they were used to differentiate between children with IBS and healthy subjects. In the case of serum omentin-1 concentrations ≤ 279.4 ng/mL, the sensitivity and specificity were 60% and 80%, respectively. Conclusions: The nutritional status of children with IBS did not differ from that of the healthy controls. We found significant differences in serum chemerin and omentin-1 concentrations between IBS patients and healthy children. These adipokines could be used as IBS biomarkers as they demonstrate good specificity and moderate sensitivity. The serum concentrations of chemerin and omentin-1 in IBS patients were related to nutritional status and insulin resistance.

## 1. Introduction

Irritable bowel syndrome (IBS) is one of the most common functional disorders of the gastrointestinal tract (GI), and it is accompanied by abdominal pain and defecation disorders, with no accompanying organic changes [1,2]. Its prevalence is increasing worldwide and is now estimated to range from 9 to 23% in the general population [3]. The etiopathogenesis of this condition remains unclear, but factors taken into consideration include motoric disorders of the GI tract, visceral hypersensitivity, disorders of the gut–brain axis, genetics, immunology, diet, and changes in the microbiome [4].

Among the above-mentioned factors, food and nutritional status are associated with symptom onset or exacerbation in a significant proportion of patients; however, the role of diet in the pathogenesis of functional gastrointestinal disorders (FGID) is poorly understood [5]. Most clinical studies have demonstrated that excessive body weight and excessive fatness are common in children with IBS [6,7,8], and obesity is a predictor of poor outcomes and disability at long-term follow-up in children with abdominal pain-related FGIDs [9]. However, recent data from a community-based cross-sectional study did not confirm these observations [10].

One of the most intensively studied hypotheses is that visceral hypersensitivity is related to immune system activation and low-grade inflammation in the walls of the intestine [2,11]. Although IBS and IBD are considered to be two different conditions, some processes may be common in both of them. Recently, studies have investigated the participation of hormones of adipose tissue in the pathogenesis of IBD. Changes in the expression of adipokines by the mesenteric adipose tissue and their serum concentrations may impact the immunological, neuroendocrine, and lymphatic systems, as well as the mucosal barrier [12,13].

In contrast to IBD, research concerning adipokines in IBS is sparse and has only examined adults. There are conflicting results in related studies, most of which have investigated leptin and adiponectin [14,15,16,17,18,19,20]. Considering the suggested relation of IBS with low-grade inflammation of the intestinal walls, disorders in adipokine expression and secretion may also occur in this condition [11,21]. Altered adipokine concentrations in IBS patients may also be related to diet and nutritional status [22,23,24,25]. Since no specific markers to help diagnose IBS have been determined thus far, adipokines may potentially serve this purpose [4,20].

For this study, four adipokines were selected according to their origin and activity: leptin, adiponectin, chemerin, and omentin-1. Leptin is a pleiotropic adipokine produced mainly in white adipose tissue (predominantly subcutaneously) but also in low quantities by the placenta, skeletal muscle, brain, P/D1 cells in the stomach, and T cells [26,27]. Apart from influencing lipid and carbohydrate homeostasis, this cytokine influences immune reactions. It stimulates B lymphocytes to produce C-reactive protein and interferon γ [12,26,27,28,29,30,31,32]. Higher leptin concentrations are associated with increased risk of inflammatory and autoimmune diseases [28,30]. More recent data on its anti-inflammatory effects suggest that leptin is an important immune modulator with a wide range of functions, playing a critical role in the pathophysiology of multiple obesity-associated diseases and infections [32].

Adiponectin is a 224 amino acid protein that is produced by adipocytes of white and brown adipose tissues. The biological role of this adipokine is multifaceted and dependent on its isoform. There are three forms of adiponectin with different molecular weights: a low-molecular-weight complex (LMW) composed of 12–18 polymer molecules, a middle-molecular-weight complex (MMW), and a high-molecular-weight complex (HMW) [23,33,34]. The polymeric forms (HMW) are associated with metabolic processes, strongly interacting in the regulation of tissue sensitivity to insulin, glucose uptake, and lipid metabolism. This form of adiponectin exerts anti-diabetic and anti-atherosclerotic effects. HMW complexes seem to activate both anti-inflammatory and pro-inflammatory signaling pathways. LMW complexes have potent anti-inflammatory bioactivity. They stimulate the production of anti-inflammatory cytokines (interleukin 10, interleukin-1 receptor antagonist) and inhibit the secretion of pro-inflammatory cytokines (IL-6, IL-8, macrophage inflammatory proteins 1γ and 1β (MIP-1γ and -1β), MCP-1, and tissue inhibitors of metalloproteinases 1 and 2 (TIMP-1 and -2)). On the other hand, globular adiponectin may have pro-inflammatory effects [29,33,34,35,36,37]. Decreased adiponectin concentrations have been found in various conditions, such as type 2 diabetes, obesity, and metabolic syndrome, while they are increased in chronic kidney disease, type 1 diabetes, lipodystrophy, anorexia nervosa, IBD, rheumatoid arthritis, and other conditions [34,35,36].

Adiponectin and leptin levels are independently associated with the development of metabolic syndrome, diabetes mellitus type II, and cardiovascular diseases. However, it has been demonstrated that the adiponectin/leptin ratio (A/L) is more strongly correlated with insulin resistance, metabolic syndrome, carotid intima-media thickness, and an “at-risk phenotype” than individual hormones. The A/L ratio can be considered a practical marker of adipose tissue dysfunction when identifying persons at increased risk of cardiometabolic diseases [38].

Chemerin is produced by adipocytes and stroma cells in adipose tissue, and it generally shows pro-inflammatory activity [39,40,41,42,43]. However, in certain circumstances, it inhibits inflammation by stimulating adiponectin production and decreasing the secretion of pro-inflammatory cytokines [40,41]. Elevated chemerin concentrations are present in chronic inflammatory diseases and in states of insulin resistance, as this adipokine has a negative impact on carbohydrate metabolism [40,41,42].

Omentin-1 is secreted by the stromal cells of visceral adipose tissue and endothelial cells [44,45]. Although its biological activity is still not fully understood, it has been shown to increase the insulin sensitivity of tissues [46] and have anti-inflammatory effects [45,47]. Decreased omentin-1 levels have been found in chronic inflammatory diseases and in states of insulin resistance [45,48,49].

The aim of this study is to assess the nutritional status and concentrations of leptin, adiponectin, omentin-1, and chemerin in children with IBS. The results were compared to those of a healthy control group. The association with clinical and laboratory parameters was also investigated.

## 2. Materials and Methods

This investigation was approved by the Bioethical Committee of the Medical University of Silesia in Katowice, Poland (KNW/0022/KB1/69/13 dated 25 June 2013). Written consent was obtained from all caregivers of the participants and all children aged 16 years or older.

### 2.1. Patients with IBS

For the study group, we enrolled 33 children (11 girls and 22 boys) aged 5–18 years who were diagnosed with IBS based on the Rome III Criteria that were in force at the time when the study was conducted. Chronic diseases were ruled out in these individuals based on their medical history and clinical and laboratory findings. They had no alarming symptoms affecting the gastrointestinal tract (weight loss, fever of an unknown cause, GI bleeding, night-time symptoms, perianal changes, prolonged vomiting, dysphagia, arthritis, growth or sexual maturation retardation, positive family history of cancer or IBD, organic disease of the GI tract). They had no acute infections within 14 days prior to the study and took no medication or diet supplements either chronically or incidentally during the month prior to the study. Pregnancy was also an exclusion criterion. These criteria were met by 33 patients (11 girls; 22 boys), of whom 26 (78.8%) had IBS with diarrhea (IBS-D), 4 (12.1%) had constipation (IBS-C), and 3 (9.1%) had alternating diarrhea and constipation (IBS-DC). The general characteristics of the group of IBS patients are shown in Table 1.

### 2.2. Control Group

The control group consisted of 30 healthy children (11 girls; 19, boys) aged 5–17 years who were recruited from patients attending the Pediatric Surgical Out-patient Clinic in Clinical Hospital No 1 in Zabrze, Poland. These individuals were seen for control visits after the treatment of minor traumas, such as joint dislocations, bruises, and superficial wounds. The exclusion criteria were the following: any chronic diseases, acute infection in the 14 days before the study, medication or ingestion of diet supplements chronically or incidentally one month before the study, and pregnancy. Data describing the control group are presented in Table 1.

### 2.3. Anthropometric Measurements

The patient and control groups were physically examined, and detailed medical history was recorded. Anthropometric measurements were conducted in the morning after fasting and urinating in a well-lit and heated room by the same person. The mean of three measurements was calculated for each parameter. Height was measured using an anthropometer (GPM, Switzerland) in a relaxed standing position, without excessive straightening and with the subject’s head positioned in the ocular-auricular plane (Frankfurt).

Weight was measured using a medical scale and used to calculate body mass index (BMI = (body weight (kg))/(height (m)^2^)). Weight, height, and BMI were expressed as standard deviation scores (SDSs) for age and gender, which were calculated using reference data published by Palczewska et al. [50].

Waist and hip circumferences were measured in a relaxed standing position using a meter after emptying the bladder. The data were used to calculate the waist-to-hip ratio (WHR = waist circumference (cm)/hip circumference (cm)) [51].

### 2.4. Laboratory Measurements

Blood was drawn for laboratory testing between 8 and 9 a.m. after 12 h of fasting. Serum samples were obtained after centrifugation (15 min, 1000× *g*, in +4 °C) and stored at −70 °C until analysis. In the case of menstruating girls, blood was drawn in the follicular phase of the menstrual cycle.

Concentrations of C-reactive protein (CRP), glucose, lipids, alanine and aspartate aminotransferases (ALT and AST), and insulin were measured in the Central Laboratory of Clinical Hospital No 1 in Zabrze using a Cobas 6000 analyzer (Cobas c 501 module; Roche Diagnostics, Basel, Switzerland). Enzyme-linked immunosorbent assays (ELISAs) were carried out in the same laboratory using commercial tests (BioVendor (Brno, Czech Republic)) to assess leptin, adiponectin, omentin-1, and chemerin concentrations. The limits of detection (LD) and the intra- and inter-assay coefficients of variability (intra-CV and inter-CV) for the sets were as follows: leptin (LD 0.2 ng/mL, intra-CV 7.6%, inter-CV 6.7%), adiponectin (LD 0.47 ng/mL, intra-CV 4.4%, inter-CV 6.2%), omentin-1 (LD 0.5 ng/mL, intra-CV 4.1%, inter-CV 4.8%), and chemerin (LD 0.1 ng/mL, intra-CV 7.0%, inter-CV 8.3%).

The homeostasis assessment model—insulin resistance (HOMA-IR) was used as the insulin resistance index and calculated based on the glucose and insulin concentrations as HOMA-IR = (insulin (μIU/mL) × glucose (mmol/L))/22.5.

### 2.5. Statistical Analysis

Statistical analyses were performed using Statistica 10.0 (StatSoft Inc., Tulsa, OK, USA). Descriptive statistics for all the variables were calculated, and adequate figures were generated. The Kołmogorow–Smirnow test was used to check the normality of distributions, and Levene’s test was used to verify the variance homogeneity of the variables. Normally distributed variables were comparatively analyzed using a Student’s t-test, ANOVA, and post hoc Tukey’s test. In cases without a distribution and a lack of variance homogeneity, the U Mann–Whitney or Kruskal–Wallis ANOVA was applied. The chi^2^ test with Yates correction was used to analyze the frequency of improper laboratory results.

Associations between two variables were estimated using Pearson’s or Spearman’s correlation coefficients depending on the distribution of the continuous variables. Values of adipokine concentrations were corrected using an analysis of covariance (ANCOVA) against clinical parameters that significantly differed between groups.

The diagnostic value of adipokine concentrations was determined using the receiver operator curves (ROCs). The sensitivity, specificity, and efficiency were calculated. The parameter limit values corresponding to the maximum test efficiency were determined for the point where sensitivity and specificity were equal.

The results were considered significant at *p* < 0.05.

## 3. Results

The patients and controls did not differ in terms of age and anthropometric parameters (Table 1). Obesity (BMI SDS > 2.0) was similarly common in both groups (15% vs. 10%; *p* = 0.87).

There were significant differences in lipid concentrations. Children with IBS had lower mean HDL cholesterol (1.26 ± 0.31 mmol/L vs. 1.49 ± 0.36 mmol/L, *p* = 0.009) and higher triglyceride (TG) (1.05 ± 0.49 mmol/L vs. 0.68 ± 0.2 mmol/L, *p* = 0.001) and HOMA-IR (2.21 ± 1.5 vs. 1.57 ± 0.81, *p* = 0.001) levels. The mean CRP, glucose, insulin, and total cholesterol concentrations and transaminase activity did not differ between the study and control groups (Table 1). There were no differences in the frequency of metabolic syndrome indicators between groups: CRP > 5 mg/L (3% vs. 3.3%, *p* = 0.51), glucose ≥ 100 mg/dL (3% vs. 6.7%, *p* = 0.96), total cholesterol > 5.2 mmol/L (9% vs. 0%, *p* = 0.54), HDL cholesterol ≤ 1.02 mmol/L (24.2% vs. 6.7%, *p* = 0.20), and TG > 1.69 (9% vs. 3.3%, *p* = 0.72).

### 3.1. Adipokine Concentrations

Serum leptin (Figure 1) and adiponectin (Figure 2) levels did not differ significantly between IBS patients and the controls. These results did not change after adjusting for gender, TG, HDL, and HOMA-IR. The L/A ratio was similar (*p* = 0.37) in IBS patients and the control group (0.82 ± 1.17 vs. 0.57 ± 0.80, respectively).

The mean chemerin concentrations were significantly higher (*p* = 0.03) in IBS patients (Figure 3); this was also found after adjusting for gender, TG, HDL, and HOMA-IR. In contrast, omentin-1 serum concentrations in IBS patients were significantly lower (*p* = 0.04) than in healthy children (Figure 4), even after adjusting for gender, TG, HDL, and HOMA-IR. Among IBS patients, leptin levels were positively correlated with chemerin concentrations (r = 0.62, *p* < 0.001) and inversely correlated with serum omentin-1 (r = −0.60, *p* < 0.001). In the control group, concentrations of adipokines showed no significant associations with each other.

### 3.2. Adipokine Concentration, Age, and Anthropometric Parameters

Age was not related to the concentration of the studied adipokines in either of the investigated groups, apart from a negative association with serum adiponectin among healthy controls (r = −0.47, *p* = 0.009). In children with IBS, leptin concentrations were positively related to weight SDS (r = 0.56, *p* < 0.001), BMI SDS (r = 0.63, *p* < 0.001), waist circumference SDS (r = 0.54, *p* = 0.001), and WHR (r = 0.33, *p* = 0.004). In healthy controls, leptin concentrations had a positive correlation only with BMI SDS (r = 0.40, *p* = 0.03) (Table 2).

Adiponectin concentrations were not significantly associated with the analyzed anthropometric parameters in either of the groups. Statistically significant correlations of serum chemerin were found only among IBS patients. The concentration of these adipokines rose with weight SDS (r = 0.39, *p* = 0.03), BMI SDS (r = 0.48, *p* = 0.007), and waist circumference SDS (r = 0.45, *p* = 0.007). In children with IBS, omentin-1 concentrations were negatively related to all anthropometric parameters: weight SDS (r = −0.43, *p* = 0.02), BMI SDS (r = −0.44, *p* = 0.02), waist circumference SDS (r = −0.44, *p* = 0.009), and WHR (r = −0.42, *p* = 0.02). In healthy children, only weight SDS (r = −0.36, *p* = 0.049) and BMI SDS (r = −0.45, *p* = 0.01) were inversely associated with serum omentin-1 (Table 2).

### 3.3. Adipokine Concentration and Laboratory Results

Significant associations were found between biochemical parameters and adipokine concentrations, predominantly observed in IBS patients. Serum leptin levels in patients and controls were positively associated with insulin concentration (r = 0.53, *p* = 0.001 and r = 0.48, *p* = 0.007, respectively) and HOMA-IR (r = 0,61, *p* < 0.001 and r = 0.39, *p* = 0.04, respectively). Moreover, in the IBS group, leptin was positively related to CRP values (r = 0.44, *p* = 0.01) and TG (r = 0.36, *p* = 0.04) (Table 3).

In both groups, adiponectin was negatively correlated with HOMA-IR values (IBS patients: r = −0.37, *p* = 0.04, controls: r = −0.44, *p* = 0.01). Additionally, in controls, adiponectin was positively related to HDL cholesterol levels (r = 0.39, *p* = 0.03) and negatively related to insulin concentration (r = −0.47, *p* = 0.009). Chemerin and omentin-1 were significantly associated with biochemical parameters only in IBS patients. Positive correlations were found between serum chemerin and total cholesterol (r = 0.45, *p* = 0.01), LDL cholesterol (r = 0.4, *p* = 0.006), TG (r = 0.4, *p* = 0.02), insulin (r = 0.5, *p* = 0.003), and HOMA-IR values (r = 0.5, *p* = 0.004). Omentin-1 was inversely associated with CRP (r = −0.36, *p* = 0.04), insulin (r = −0.5, *p* = 0.005), TG (r = −0.3, *p* = 0.04), and HOMA-IR values (r = −0.40, *p* = 0.006) and positively associated with HDL cholesterol levels only (r = 0.38, *p* = 0.04) (Table 3).

### 3.4. ROC Analysis

There were significant differences in chemerin and omentin-1 between IBS patients and controls. ROC plots were used to determine cut-off points for the serum levels of these two adipokines in IBS.

Chemerin concentrations ≥ 232.8 ng/mL showed quite low sensitivity (39%), but good specificity (87%) in differentiating patients with IBS from health controls (Figure 5). Omentin-1 levels ≤ 279.4 ng/mL had 60% sensitivity and 80% specificity (Figure 6).

## 4. Discussion

The most important findings of the present study were the significantly higher chemerin and loweromentin-1 concentrations in children with IBS compared to controls, as well as the lack of differences in adiponectin and leptin levels between these two groups. As of the time of writing, there are no published data on the hormonal activity of adipose tissue in children with IBS, and not much research has been conducted in adults. Moreover, the results of the sparse studies available are conflicting [14,15,16,17,18,19,20]. To our knowledge, there are no published data on the use of adipokines as markers of IBS. Our results might point to a new path in the search for laboratory parameters for IBS diagnostics using adipose tissue hormones.

Leptin concentrations did not differ between children with IBS and healthy controls. We demonstrated their positive correlation with BMI, weight, and waist circumference, as well as the metabolic parameters HOMA-IR, CRP, TG, and insulin levels. One study conducted in adults with IBS revealed significantly higher leptin concentrations compared to healthy control subjects [14], while another study found lower levels [15]. The findings of most authors [16,18,19] are in accordance with our results. The studies mentioned [14,15,18,19] confirmed the well-known association between leptin and weight/BMI [14,27,52], but metabolic parameters were generally not assessed, except in the research by Weaver et al. [18], who demonstrated a positive correlation between leptin and total cholesterol in females. Although Russo et al. [16] did not find any significant differences in leptin levels between patients with diarrhea-predominant IBS (IBS-D) and healthy controls, their more thorough research demonstrated significantly higher leptin concentrations only in the IBS-D subgroup with altered small-intestinal permeability [17]. Additionally, sex hormones may modulate leptin secretion directly and indirectly by impacting the distribution of adipose tissue in the body [53]. Interestingly, Weaver et al. [18] postulate that disturbances in the functioning of normal feedback mechanisms along the hypothalamic–pituitary–gonadal axis observed in women with IBS may be involved in the exacerbation of symptoms.

In our study, there were also no differences in adiponectin concentrations between the groups. There are no data on serum adiponectin levels in children with IBS, and studies involving adult patients have yielded contradictory findings [16,17,20]. In the two publications by Russo et al. [16,17], serum adiponectin levels were significantly higher in patients with IBS-D (especially with normal small-intestinal permeability) in comparison to healthy controls, whereas Baram et al. [20] reported the opposite results in a group including patients with all types of IBS. Unfortunately, correlations between adiponectin and anthropometric and metabolic parameters were not assessed in these studies, and the clinical characteristics of the examined groups only involved BMI [16,17,20] and body fat percentage [20], which were not different between patients and healthy controls.

Similarly to Baram et al. [20], we found significantly higher chemerin concentrations in IBS patients than in controls. They also demonstrated that an increase in pro-inflammatory adipokine levels is associated with an increase in the severity of symptoms [20].

Interestingly, in our study, correlations between chemerin, anthropometric, and biochemical parameters and concentrations of other adipokines were present only among children with IBS who had higher chemerin levels than the controls. This might suggest that such relations develop in non-physiological states. Similar results were shown in a meta-analysis concerning metabolic syndrome [54]. Such observations were also noted in children with obesity [55]. In addition, an association between chemerin and insulin resistance has been found in various studies on children and adults who are healthy, obese, or have type 2 diabetes [40,56,57,58,59]. Therefore, higher chemerin levels in the IBS group should be considered as an adverse factor that may be related to the metabolic disorders observed in this group. Higher chemerin was also found in adults with IBD than in healthy controls, especially in men [60,61]; animal studies have indicated that it may take part in the development of bowel inflammation and could also experimentally induce it. Thus, chemerin is suggested as a biomarker of the inflammation severity in IBD and a potential therapeutic target [62].

In our group of children with IBS, omentin-1 levels were significantly lower than in healthy controls, and they correlated negatively with anthropometric and biochemical parameters. Unfortunately, no data on serum omentin-1 in patients with IBS have been published so far. We can only speculate that, due to omentin-1′s anti-inflammatory activity, omentin-1 concentrations may inversely reflect the severity of low-grade inflammation. It may also be suggested that decreased serum omentin-1 in IBS patients could be involved in the metabolic disorders observed in this group, as this adipokine has been proven to have a positive impact on insulin sensitivity and the concentrations of adiponectin and HDL [63,64,65].

The varying conclusions concerning adipokine concentration in IBS patients may result from several factors that influence the secretion of these hormones, such as the mass of fat tissue and its distribution, age, insulin sensitivity, IBS type and duration, medication, sex hormones, stress, and the presence of low-grade inflammation [4,12,21,29]. The latter component is a relevant issue in the pathophysiology of IBS [4]. Briefly, the greater magnitude of gut microbiota dysbiosis in patients with IBS demonstrated in many studies may contribute to increased intestinal permeability and consequent barrier dysfunction that allows the permeation of pathogenic microorganisms and their products. Furthermore, these infections lead to the increment of various immune-related cells, such as macrophages, mast cells, T-lymphocytes, and pro-inflammatory cytokines (TNF-α, IFN-γ, IL-3, IL-4, IL-5, and IL-6), resulting in low-grade inflammation that has a notable effect on vascular permeability, gastrointestinal motility, secretion, and visceral hypersensitivity, as these play key roles in the development of IBS symptoms [4,66]. Low-grade inflammation in visceral adipose tissue may also potentially result in alterations in adipokine secretion [29]. Therefore, it could be speculated that higher chemerin—taking part in the generation and maintenance of inflammation and lower anti-inflammatory omentin-1 levels—may contribute to the pathophysiology of IBS. These specific adipokines were also found to be specific markers for differentiating IBS from healthy children, although their sensitivity was low or moderate. Since IBS symptoms also occur in many other diseases, a specific and sensitive marker for IBS would certainly make the diagnosis easier and more cost effective [67,68].

An interesting aspect of our study is metabolic disorders in children with IBS. Our results may indicate an increased risk for developing metabolic syndrome in IBS patients, which has been suggested in other publications [69,70]. Indicators of metabolic syndrome, such as higher triglycerides and HOMA-IR, as well as lower HDL cholesterol, were revealed in children with IBS, despite a lack of differences in anthropometric measurements. However, one should be aware that anthropometric parameters are not sensitive enough in the early diagnosis of metabolic disorders. The adverse changes in gut microbiome, intestinal low-grade inflammation, and oxidative stress in IBS may lead to visceral adipose tissue dysfunction manifested by an imbalance between pro-inflammatory (e.g., chemerin) and anti-inflammatory (e.g., omentin-1) adipokines. Impaired function of adipose tissue and increased triglyceride absorption in IBS may promote ectopic lipid accumulation in the liver, skeletal muscle tissue, and around the heart. Such abnormal fat content can be visualized on an MRI long before obvious physical changes appear [70]. Adipose tissue may also accumulate around small mesenteric lymphatic and blood vessels due to their increased permeability caused by low-grade inflammation [71]. Our observations are consistent with this theory since only adipokines produced by the adipose tissue stroma differed significantly between children with IBS and controls. Deposition of fat in non-adipose tissue may be associated with insulin resistance and metabolic complications of obesity (lipotoxicity). Whether insulin resistance is a cause or consequence of metabolic inflexibility—i.e., the impaired ability of the body to switch from fat to carbohydrate oxidation—remains unresolved [72]. The metabolically “inflexible” state is typically characterized by decreased fat oxidation during fasting and a reduced ability to upregulate carbohydrate oxidation during feeding [73]. This impaired fuel switching in mitochondria may result from the continuous influx of fuel sources due to overnutrition, as well as physical inactivity and sedentary behaviors [72,73]. We did not assess the subjects’ physical activity and fitness levels; however, possible sedentary lifestyle and avoiding exercises due to somatic (abdominal pain, diarrhea) and mental (anxiety and mood disorders) stresses caused by IBS may have potentially affected the obtained results. In endurance-trained subjects, there are many structural and functional changes in muscle fibers affecting intramyocellular triacylglycerol depletion and storage. Such muscle, despite having elevated lipid content, remains markedly insulin-sensitive [72]. Recent data confirm that the most active children have better anthropometric, metabolic, and inflammatory profiles, which may lead to a better cardiometabolic status [74]. Physical activity can also stimulate variations in the gut microbiota through numerous mechanisms, such as myokine release, increased intestinal transit, or the secretion of neurotransmitters and hormones, and it can potentially modify the IBS course [75]. Therefore, further research on adipokine levels in IBS patients should also include physical activity.

This investigation has some limitations, such as its cross-sectional design and relatively small study groups. We did not analyze the socioeconomic status of the subjects, which may have affected their nutrition and their physical activity and fitness levels. Nonetheless, our research is original and there are very little data concerning adipokines in IBS, especially for chemerin and omentin-1. Thus, the presented results could lead to new hypotheses and starting points for further investigations.

## 5. Conclusions

The concentrations of adipokines secreted by stromal cells of visceral adipose tissue (chemerin and omentin-1) showed significant differences in children with IBS in comparison to healthy subjects. At the same time, there was no discrepancy in serum levels of hormones mainly produced by adipocytes. The observed changes indirectly suggest the potential participation of visceral adipose tissue in the pathogenesis of IBS.

The adipokine concentrations in the investigated children were related to nutritional status and, in the case of chemerin and omentin-1, also to insulin resistance. Therefore, their altered concentrations may contribute to the development of metabolic disorders in IBS patients.

Chemerin and omentin-1 might be considered as IBS biomarkers with good specificity and moderate sensitivity.

## Figures and Tables

**Figure 1 nutrients-14-05282-f001:**
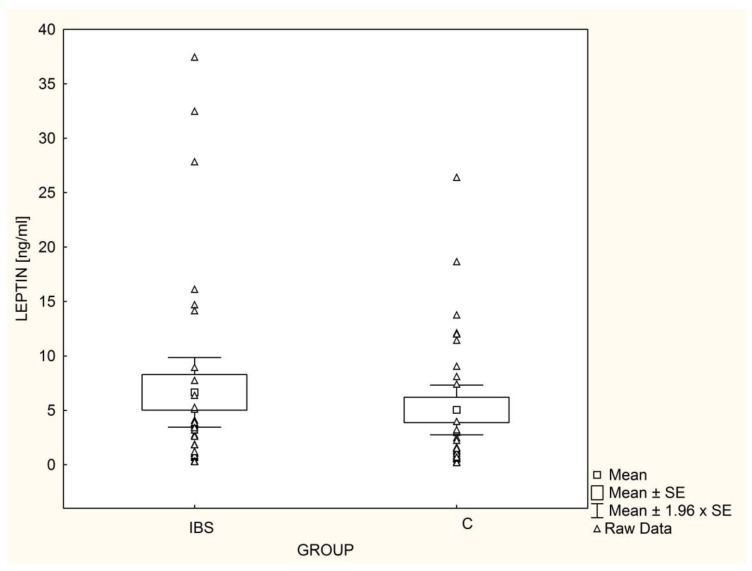
Mean leptin levels in children with irritable bowel syndrome (IBS; n = 33) and controls (C; n = 30).

**Figure 2 nutrients-14-05282-f002:**
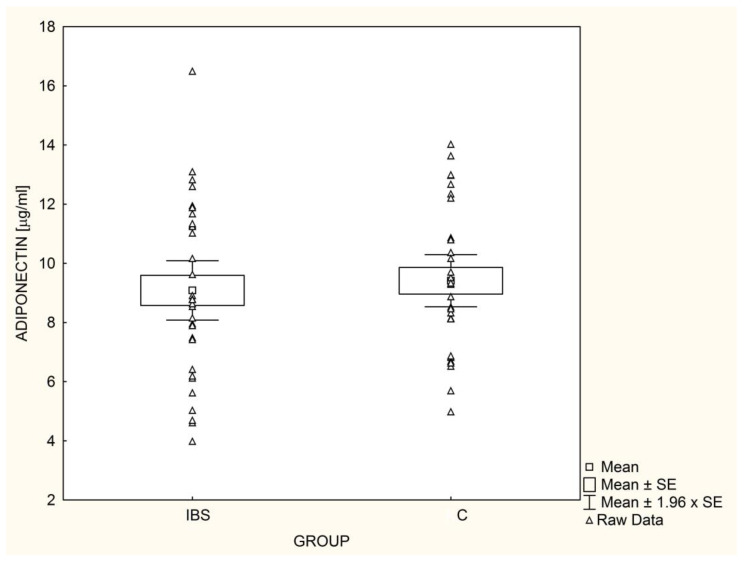
Mean adiponectin levels in children with irritable bowel syndrome (IBS; n = 33) and controls (C; n = 30).

**Figure 3 nutrients-14-05282-f003:**
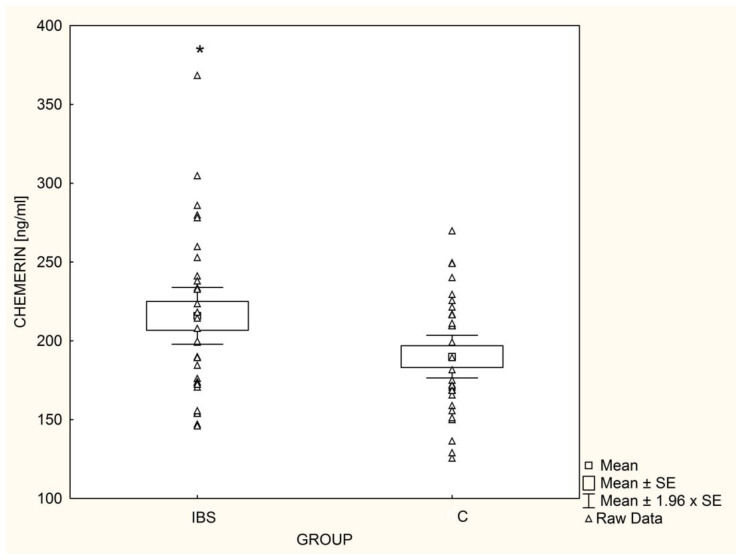
Mean chemerin levels in children with irritable bowel syndrome (IBS; n = 33) and controls (C; n = 30), * *p* = 0.03.

**Figure 4 nutrients-14-05282-f004:**
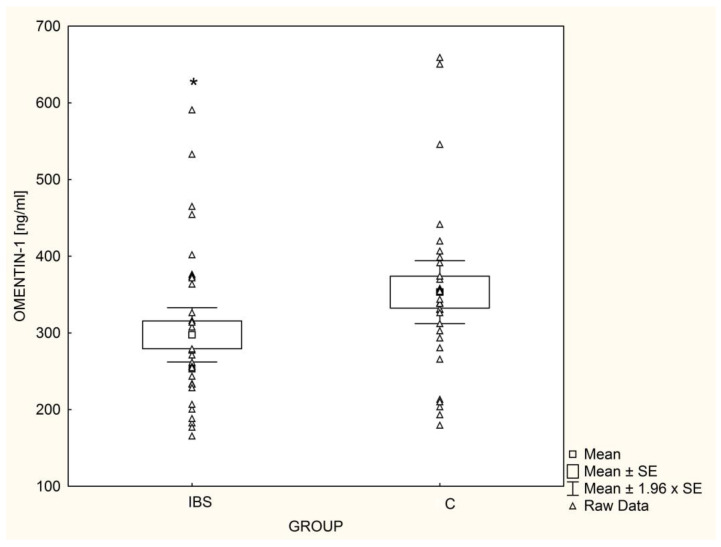
Mean omentin-1 levels in children with irritable bowel syndrome (IBS; n = 33) and controls (C; n = 30), * *p* = 0.04.

**Figure 5 nutrients-14-05282-f005:**
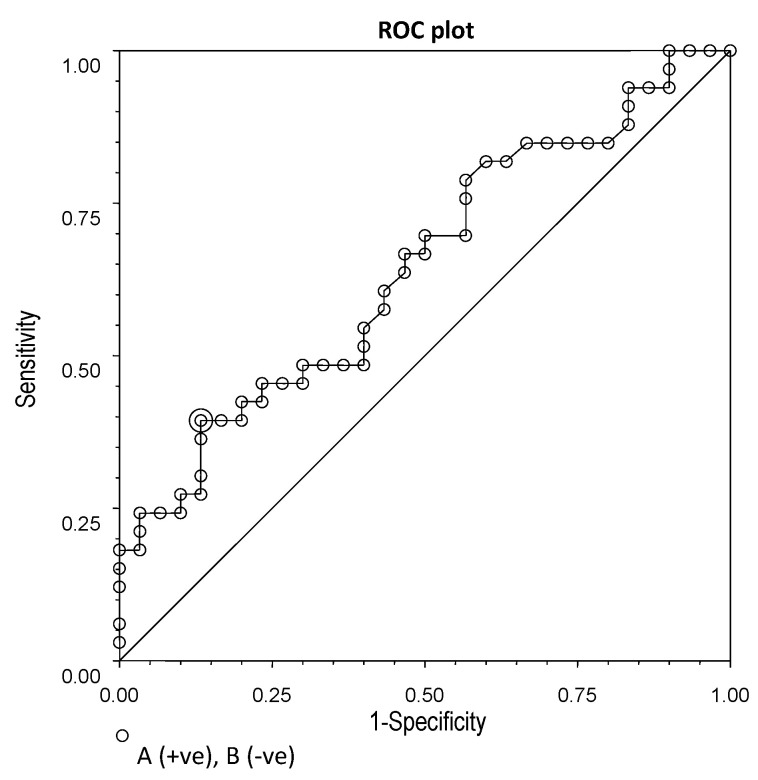
ROC plot for chemerin concentrations (cut-off point: 232.8 ng/mL; sensitivity: 0.39; specificity: 0.87; efficiency: 0.76).

**Figure 6 nutrients-14-05282-f006:**
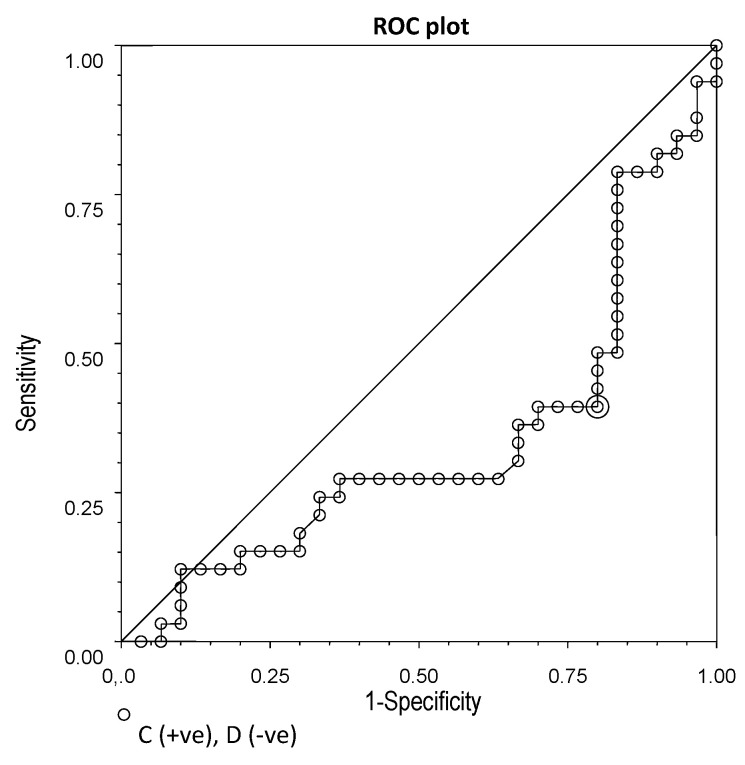
ROC plot for chemerin concentrations (cut-off point: 232.8 ng/mL; sensitivity: 0.60; specificity: 0.80; efficiency: 0.64).

**Table 1 nutrients-14-05282-t001:** Clinical characteristics of the irritable bowel syndrome (IBS) patients and control group (C). Data are presented as means, standard deviations, and ranges.

Parameter	Group	
	IBS (n = 33)	Controls (n = 30)
Age (years)	13.6 ± 3.2 (5.7–17.8)	13.6 ± 3.4 (5.1–17.5)
Height (cm)	163.2 ± 18.5 (119.5–185.0)	161.6 ± 19.7 (118.9–187.9)
Weight (kg)	56.4 ± 21.4 (23.4–110.0)	52.6 ± 19.5 (20.2–90.9)
Weight SDS	0.47 ± 1.87 (−2.13–4.67)	0.05 ± 1.14 (−1.42–2.58)
BMI (kg/m^2^)	20.00 ± 4.69 (12.64–30.09)	19.36 ± 4.31 (14.29–30.87)
BMI SDS	0.22 ± 1.71 (−2.2–4.48)	−0.09 ± 1.33 (−1.38–3.66)
Waist circumference (cm)	69.4 ± 12.4 (50.3–92.6)	67.1 ± 10.4 (51.2–93.2)
Waist circumference SDS	0.58 ± 2.75 (−3.07–6.93)	0.19 ± 1.77 (−1.88–4.32)
WHR	0.82 ± 0.05 (0.74–0.94)	0.80 ± 0.05 (0.68–0.91)
CRP (mg/L)	1.19 ± 1.28 (0.0–6.2)	0.94 ± 0.84 (0.0–3.6)
Glucose (mg/dL)	89.6 ± 6.9 (77.0–104.0)	85.8 ± 8.0 (65.0–102.0)
Insulin (μU/mL)	9.98 ± 5.99 (1.62–29.52)	7.92 ± 4.53 (1.13–20.97)
ALT (U/L)	9.8 ± 5.0 (5.8–29.1)	10.2 ± 4.0 (4.3–22.9)
AST (U/L)	18.2 ± 4.59 (10.7–32.3)	19.8 ± 5.7 (11.8–36.0)
Total cholesterol (mmol/L)	3.92 ± 0.82 (2.67–6.16)	3.83 ± 0.58 (2.63–4.97)
HDL cholesterol (mmol/L)	1.26 ± 0.31 ^a^ (0.86–2.17)	1.49 ± 0.36 (0.92–2.29)
LDL cholesterol (mmol/L)	2.18 ± 0.68 (1.11–4.27)	2.02 ± 0.45 (1.06–2.83)
Triglycerides (mmol/L)	1.05 ± 0.49 ^b^ (0.37–2.81)	0.68 ± 0.32 (0.42–1.94)
HOMA-IR	2.21 ± 1.5 ^b^ (0.36–7.14)	1.57 ± 0.81 (0.23–3.59)

BMI—body mass index; WHR—waist-to-hip ratio; SDS—standard deviation score; CRP—C-reactive protein; ALT—alanine aminotransferase; AST—aspartate aminotransferase; HOMA-IR—insulin resistance ratio; ^a^
*p* = 0.009 patients with IBS vs. controls; ^b^
*p* = 0.001 patients with IBS vs. controls.

**Table 2 nutrients-14-05282-t002:** Analysis of correlations between examined adipokines and anthropometric data.

Parameter	IBS (n = 33)				Controls (n = 30)			
	Leptin (ng/mL)	Adiponectin (µg/mL)	Chemerin (ng/mL)	Omentin-1 (ng/mL)	Leptin (ng/mL)	Adiponectin (µg/mL)	Chemerin (ng/mL)	Omentin-1 (ng/mL)
Age (years)	r = −0.07 *p* = 0.70	r = −0.34 *p* = 0.06	r = 0.11 *p* = 0.58	r = −0.22 *p* = 0.24	r = 0.27 *p* = 0.15	r = −0.47 * *p* = 0.009	r = −0.31 *p* = 0.09	r = 0.14 *p* = 0.46
Height (cm)	r = −0.01 *p* = 0.99	r = −0.49 * *p* = 0.03	r = 0.09 *p* = 0.63	r = −0.39 * *p* = 0.03	r = −0.04 *p* = 0.84	r = −0.48 * *p* = 0.007	r = −0.52 * *p* = 0.004	r = 0.10 *p* = 0.59
Weight (kg)	r = 0.44 * *p* = 0.01	r = −0.51 * *p* = 0.004	r = 0.33 *p* = 0.08	r = −0.49 * *p* = 0.006	r = 0.31 *p* = 0.09	r = −0.58 * *p* = 0.001	r = −0.34 *p* = 0.06	r = −0.14 *p* = 0.47
Weight SDS	r = 0.56 * *p* < 0.001	r = −0.31 *p* = 0.10	r = 0.39 *p* = 0.03	r = −0.43 * *p* = 0.02	r = 0.27 *p* = 0.15	r = −0.28 *p* = 0.13	r = −0.16 *p* = 0.40	r = −0.36 * *p* = 0.049
BMI (kg/m^2^)	r = 0.57 * *p* < 0.001	r = −0.41 * *p* = 0.02	r = 0.47 *p* = 0.01	r = −0.52 * *p* = 0.003	r = 0.44 *p* = 0.01	r = −0.49 * *p* = 0.006	r = −0.10 *p* = 0.59	r = −0.31 *p* = 0.10
BMI-SDS	r = 0.63 * *p* < 0.001	r = −0.25 *p* = 0.10	r = 0.48 *p* = 0.007	r = −0.44 * *p* = 0.02	r = 0.40 *p* = 0.03	r = −0.33 *p* = 0.08	r = −0.02 *p* = 0.90	r = −0.45 * *p* = 0.01
Waist circ. (cm)	r = 0.49 * *p* < 0.001	r = −0.52 * *p* = 0.003	r = 0.36 *p* = 0.05	r = −0.54 * *p* = 0.002	r = 0.39 *p* = 0.03	r = −0.48 * *p* = 0.007	r = −0.28 *p* = 0.13	r = −0.12 *p* = 0.50
Waist circ. SDS	r = 0.54 * *p* = 0.001	r = −0.26 *p* = 0.14	r = 0.45 *p* = 0.007	r = −0.44 * *p* = 0.009	r = 0.35 *p* = 0.05	r = −0.25 *p* = 0.18	r = −0.01 *p* = 0.93	r = −0.21 *p* = 0.26
WHR	r = 0.33 * *p* = 0.004	r = −0.25 *p* = 0.17	r = 0.26 *p* = 0.16	r = −0.42 * *p* = 0.02	r = −0.06 *p* = 0.76	r = 0.80 *p* = 0.69	r = 0.04 *p* = 0.84	r = −0.17 *p* = 0.37

IBS—irritable bowel syndrome; BMI—body mass index; WHR—waist-to-hip ratio; SDS—standard deviation score; * statistically significant.

**Table 3 nutrients-14-05282-t003:** Analysis of correlations between examined adipokines and laboratory results.

Parameter	IBS (n = 33)				Controls (n = 30)			
	Leptin (ng/mL)	Adiponectin (µg/mL)	Chemerin (ng/mL)	Omentin-1 (ng/mL)	Leptin (ng/mL)	Adiponectin (µg/mL)	Chemerin (ng/mL)	Omentin-1 (ng/mL)
CRP (mg/L)	r = 0.44 * *p* = 0.01	r = −0.06 *p* = 0.75	r = 0.31 *p* = 0.08	r = −0.36 * *p* = 0.04	r = −0.06 *p* = 0.77	r = −0.07 *p* = 0.71	r = 0.33 *p* = 0.08	r = 0.13 *p* = 0.50
Glucose (mg/dL)	r = 0.29 *p* = 0.12	r = −0.15 *p* = 0.44	r = 0.10 *p* = 0.62	r = −0.30 *p* = 0.11	r = −0.07 *p* = 0.70	r = 0.05 *p* = 0.80	r = −0.25 *p* = 0.80	r = −0.29 *p* = 0.12
Insulin (μU/mL)	r = 0.53 * *p* = 0.001	r = −0.36 *p* = 0.05	r = 0.53 * *p* = 0.003	r = −0.50 * *p* = 0.005	r = 0.48 * *p* = 0.007	r = −0.47 * *p* = 0.009	r = −0.15 *p* = 0.42	r = −0.30 *p* = 0.11
ALT (U/L)	r = 0.30 *p* = 0.90	r = −0.15 *p* = 0.42	r = 0.18 *p* = 0.31	r = −0.21 *p* = 0.24	r = 0.23 *p* = 0.22	r = −0.11 *p* = 0.58	r = −0.34 *p* = 0.07	r = −0.34 *p* = 0.07
AST (U/L)	r = −0.17 *p* = 0.35	r = −0.03 *p* = 0.90	r = −0.04 *p* = 0.82	r = 0.26 *p* = 0.16	r = −0.31 *p* = 0.09	r = 0.16 *p* = 0.39	r = −0.23 *p* = 0.22	r = −0.09 *p* = 0.65
Total cholesterol (mmol/L)	r = 0.21 *p* = 0.24	r = 0.37 * *p* = 0.05	r = 0.45 * *p* = 0.01	r = 0.06 *p* = 0.74	r = −0.03 *p* = 0.89	r = 0.29 *p* = 0.12	r = −0.06 *p* = 0.77	r = −0.16 *p* = 0.40
HDL cholesterol (mmol/L)	r = −0.19 *p* = 0.28	r = 0.19 *p* = 0.31	r = −0.17 *p* = 0.37	r = 0.38 * *p* = 0.04	r = 0.01 *p* = 0.99	r = 0.39 * *p* = 0.03	r = −0.10 *p* = 0.60	r = −0.10 *p* = 0.60
LDL cholesterol (mmol/L)	r = 0.26 *p* = 0.14	r = 0.36 * *p* = 0.05	r = 0.49 * *p* = 0.006	r = 0.03 *p* = 0.88	r = −0.03 *p* = 0.86	r = 0.17 *p* = 0.37	r = −0.01 *p* = 0.98	r = −0.08 *p* = 0.68
Triglycerides (mmol/L)	r = 0.36 * *p* = 0.04	r = 0.03 *p* = 0.86	r = 0.41 * *p* = 0.02	r = −0.37 * *p* = 0.04	r = 0.18 *p* = 0.34	r = −0.33 *p* = 0.08	r = 0.07 *p* = 0.70	r = −0.11 *p* = 0.57
HOMA-IR	r = 0.61 * *p* < 0.001	r = −0.37 * *p* = 0.04	r = 0.50 * *p* = 0.004	r = −0.49 * *p* = 0.006	r = 0.39 * *p* = 0.04	r = −0.44 * *p* = 0.01	r = −0.23 *p* = 0.21	r = −0.26 *p* = 0.21

IBS—irritable bowel syndrome; CRP—C-reactive protein; ALT—alanine aminotransferase; AST—aspartate aminotransferase; HOMA-IR—insulin resistance ratio; * statistically significant.

## Data Availability

The data presented in this study are available on request from the corresponding author.

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
