# Peer review of "Nutritional Status and Selected Adipokines in Children with Irritable Bowel Syndrome"

_nutrients, 2022, doi:10.3390/nu14245282_

Round 1

Reviewer 1 Report

This is overall a nice little study that certainly adds to our understanding of IBS. However, in its present form, the paper does require a little more work in terms of the English grammar, which is way off in places, so need a good English editor to go through it and could probably be made a bit stronger by a broader discussion around the underlying causes of the metabolic syndrome and their relationship to physical activity. Specifics:

1)      Abstract. Wrong use of semi-colons.

2)      Line 27. Very difficult to read sentence. Rewrite.

3)      Lines 49, 54, 65, 71. Poor English.

4)       Line 74. “B lymphocytes”

5)      Line 79. Which forms of adiponectin (it is quite complex)?

6)      Line 87. Poor English.

7)      Overall, perhaps ought to mention that the metabolic syndrome is caused as much, if not more, by a lack of physical activity, when compared to poor nutrition.

8)      Line 108. Rome III criteria? (Written around the wrong way?)

9)      Line 114. Poor English.

10)  Line 147. Superscript missing on temperatures.

11)  Line 243. These are all indicators of the metabolic syndrome.

12)  Line 299. Any point in looking at the significance of the adiponectin to leptin ratio?

13)  Line 426. Conclusion differs from elsewhere (back to front)!

14)  Line 435. May need to mention that leptin can be releases by muscle and can have both inflammatory and anti-inflammatory actions.

15)  Line 485. Perhaps ought to emphasise the link between the metabolic syndrome and inflammation. The key point here is that the dyslipidaemia and insulin resistance are all hallmarks of the acute phase response, as is the CRP level. All linked to metabolic inflexibility.

16)  Line 494. Key here is a possible link to a shift in the gut microbiota and its leakage, all linked to a shift in inflammatory tone. Plenty of data around this.

17)  Line 508. Must remember that the anthropometric biomarkers the authors have used are not that sensitive. For example, changes in fat distribution, and fat in say, the liver and muscle, and around the heart, can be picked up by MRI long before more obvious physical changes are seen. Look up the TOFI concept. BMI is not really a good indicator. For example, it is quite possible to have a BMI of 30 or more, but not have any metabolic problems at all; just look at most of the Rugby/American football teams. Even Sumo wrestlers can have relatively normal metabolism. What is perhaps also interesting is that in this population of children, the BMI is actually very low. However, the metabolic markers are telling a different story.

18)  Overall, the one thing that is missing in this discussion is the role of physical activity and fitness. Although the authors have not measured it, it could play a key role in the phenotype they are observing, so it must be discussed. Muscle is critical in modulating inflammation. Studying it would thus make a valuable follow up study. We know that exercise can dramatically affect the microbiota as well. What about the relationship to myokines?

Author Response

Thank you very much for your thorough review and valuable comments.

1, 4, 8, 10, 13. Corrected.

2, 3, 6, 9. The manuscript has been corrected by the MDPI English editor (certificate attached).

5. Information on the adiponectin isoforms and their biological roles has been added (lines 85-97).

7, 18.  The role of physical activity and fitness in the metabolic syndrome pathogenesis and their possible effect on the results of the study have been discussed (lines 544-556).

12. The leptin/adiponectin ratio has been calculated. There were no significant differences between the groups. Information on usefulness of L/A ratio has been added in the Introduction (lines 101-107) and the calculated values for examined groups in lines 242-243.

14. Information has been added in lines 73-75 and 79-82.

15, 17. The link between metabolic syndrome and inflammation and limited sensitivity of anthropometric measurements have been discussed in lines 521-532.

16. The link between gut microbiota, its leakage and shift in inflammatory tone has been provided (lines 502-510).

Reviewer 2 Report

This research article by Roczniak, W. et. al., assessed the nutritional status and concentrations of leptin, adiponectin, omentin-1, and chemerin in children with IBS. The results were compared to those of the healthy control group. The observed changes indirectly suggest the potential participation of visceral adipose tissue in the pathogenesis of IBS. This is a very simple yet interesting study that is nicely conducted. I have a few minor comments/suggestions.

1. Line 71: Leptin is mainly produced.

2. Did the authors measure pro- and anti-inflammatory markers in the stool samples from IBS and control group patients?

3. Do the authors know about the socio-economic background of the participants of this study (both IBS and control group)? This information might shed some light on the nutritional habits of the participants. 

Author Response

Thank you very much for your valuable suggestions.

  1. Corrected.
  2. Fecal calprotectin was evaluated in IBS patients and gave negative results.  In 24 from 33 patients with IBS colonoscopy was performed which did not reveal pathological changes.
  3. The socio-economic background of the participants of this study has not been evaluated, however we could assume that it was similar. Both IBS and controls were patients of the same hospital providing health service in the same area.